# Effect of Post-Harvest LED and UV Light Irradiation on the Accumulation of Flavonoids and Limonoids in the Segments of Newhall Navel Oranges (*Citrus sinensis* Osbeck)

**DOI:** 10.3390/molecules24091755

**Published:** 2019-05-06

**Authors:** Shengyu Liu, Linping Hu, Dong Jiang, Wanpeng Xi

**Affiliations:** 1College of Horticulture and Landscape Architecture, Southwest University, Chongqing 400716, China; liushengyu2018@163.com (S.L.); 18875065318@163.com (L.H.); 2Citrus Research Institute, Chinese Academy of Agricultural Sciences, Chongqing 400712, China; jiangdong@cric.cn; 3Key Laboratory of Horticulture Science for Southern Mountainous Regions, Ministry of Education, Chongqing 400715, China

**Keywords:** Newhall navel oranges, LED, UVs, PMFs, flavonoid glucosides, limonoids

## Abstract

To investigate the effect of post-harvest light irradiation on the accumulation of flavonoids and limonoids, harvested Newhall navel oranges were continuously exposed to light-emitting diode (LED) and ultraviolet (UV) light irradiation for 6 days, and the composition and content of flavonoids and limonoids in the segments were determined using UPLC-qTOF-MS at 0, 6, and 15 days after harvest. In total, six polymethoxylated flavonoids (PMFs), five flavone-*O*/*C*-glycosides, seven flavanone-*O*-glycosides, and three limonoids were identified in the segments. The accumulation of these components was altered by light irradiation. Red and blue light resulted in higher levels of PMFs during exposure periods. The accumulation of PMFs was also significantly induced after white light, UVB and UVC irradiation were removed. Red and UVC irradiation induced the accumulation of flavone and flavanone glycosides throughout the entire experimental period. Single light induced limonoid accumulation during exposure periods, but limonoid levels decreased significantly when irradiation was removed. Principal component analysis showed a clear correlation between PMFs and white light, between flavonoid glycosides and red light and UVC, and between limonoids and UVC. These results suggest that the accumulation of flavonoids and limonoids in citrus is regulated by light irradiation. White light, red light and UVC irradiation might be a good potential method for improving the nutrition and flavor quality of post-harvest citrus.

## 1. Introduction

Many physical or chemical methods, such as cold storage [1], heat treatment [2], light irradiation [3], and natural antibacterial agents [4] are used to delay the loss of fruit quality after harvest. Post-harvest chemical treatments for preserving fruit are becoming less accepted by consumers due to the possible pollution or other undesired residues. Light irradiation has attracted increasing attention due to its advantages of being highly efficient and residue free as well as its ability to control decay and extend storage life. It has been reported that exposure to UVC and UVB induces large increases in resveratrol derivatives in post-harvest grapes [5]. Traditionally, UVC irradiation has been recognized as an effective sterilization method to increase the shelf-life of fruits [6,7]. Continuous white–blue light exposure increases the storage time of broccoli and contributes to a higher accumulation of antioxidant compounds [8]. These findings suggest that both ultraviolet and other lights have the potential to be applied for maintaining or enriching the health-promoting components of post-harvest fruits.

Previous studies have investigated the effect of red and blue LED light irradiation on β-cryptoxanthin accumulation in the flavedo of Satsuma mandarin (*Citrus unshiu* Marc.), the combination of ethylene and red LED light irradiation on carotenoid metabolism [9], and the regulation role of red and blue LED light irradiation in ascorbic acid metabolism in citrus juice sacs of Satsuma mandarin (*Citrus unshiu* Marc.), Valencia orange (*C. sinensis* Osbeck), and Lisbon lemon (*C. limon* Burm.f.) [10]. To date, the effect of light irradiation on flavonoid metabolism in Newhall navel oranges or other citrus fruits has not yet been reported. Though similar studies were conducted in pear, apple, and mango, only total flavonoid content was determined, or single light irradiation was used. A global evaluation of multiple light sources regarding bioactive compounds in fruit is still lacking.

Flavonoids and limonoids are two major groups of health-promoting components in citrus fruits that exhibit a broad range of biological activities, such as antioxidant [11], anti-proliferative [12], anti-atherosclerotic [13], and neuroprotective effects [14]. Light treatment reportedly increases total flavonoid content in a wide range of species, including pear [15], apple [16], mango [6], and Brussel sprouts [17]. However, the majority of studies have only examined the impact of UVB or blue light treatment on health-promoting components in fruits, and less information is available for other lights on this subject.

Newhall navel orange (*Citrus sinensis* Osbeck) is one of the most important orange cultivars distributed worldwide for its special flavor and good performance during storage [18]. The fruits have high nutritional value, and are abundant in hesperidin, narirutin, and limonin. These compounds present health-promoting properties such as antioxidant, anticancer, anti-atherosclerotic, and vasodilatation effects [19,20]. Due to its high overall quality, Newhall navel orange has gradually become one of the leading fresh citrus cultivars in China [18,21]. Currently, China is the largest producers of Newhall navel orange in the world. By the end of 2018, the total planting area in China had reached around 2000 thousand hectares, with an annual output of approximately 4 million tons, and play an important role in citrus production in China. In this work, we investigated the effect of both UV and LED irradiation on individual flavonoids and limonoids in the edible portion (segments) of Newhall navel orange to determine the optimal light quality for maintaining or improving nutritional quality during storage.

## 2. Results

### 2.1. Identification of Flavonoids and Limonoids

From 2–16 min of retention time, every ion peak was analyzed in both low- and high-energy modes (Appendix A). Each identified compound exhibited extremely similar retention times, accurate mass, and MS^2^ ion fragments with reported information for the same compound (Table 1 and Appendix A) [19]. For example, in the first observed flavonoid, the ion peak was found at 3.36 min with the quasi-molecular ion [M + H]^+^ at *m*/*z* 595.1659 in the low-energy mode (Figure 1a), corresponding to the reported retention time at 3.44 min with the MS ion at *m*/*z* 595.1652 (Appendix A). Through elementary composition matching, the molecular formula was identified as C_27_H_30_O_15_. In addition, in the high-energy mode, prominent MS^2^ ion fragments were observed at *m*/*z* 595.16565, 325.07041, 457.11183, 379.08087, 409.09118, and 477.11946 (Figure 1b), corresponding to the reported diagnostic ions at *m*/*z* 595.16484 [M + H]^+^, 325.06996 [(M + H-^0,2^X-150)]^+^, 457.11142 [(M + H)-^0,2^X-H_2_O]^+^, 379.08057 [(M + H)-^0,2^X-96]^+^, 409.09124 [(M + H)-150-2·H^2^O]^+^ and 477.12020 [^1,3^A^+^] (Appendix A). Therefore, the first compound was identified as apigenin-6,8-di-*C*-glucoside (vicenin-2). In total, twenty-one health-promoting phytochemicals were identified in the segments of Newhall navel oranges, including six polymethoxylated flavonoids, five flavone-*O*/*C*-glycosides, seven flavanone-*O*-glycosides, and three limonoids (Table 1).

After identification, the peak area of [M + H]^+^ for each compound was integrally computed to acquire MS response values (Appendix A), which were directly used to represent the content of each compound without further quantification.

### 2.2. Effect of Light Irradiation on Flavonoids and Limonoids in the Segments of Newhall Navel Oranges

Six PMFs, namely, 5,7,8,3′,4′-pentamethoxyflavone (isosinensetin), 5,6,7,8,4′-pentamethoxyflavone (tangeretin), 5,7,8,4′-tetramethoxyflavone, 5,6,7,8,3′,4′-hexamethoxyflavone (nobiletin), 5,6,7,3′,4′-pentamethoxyflavone (sinensetin), and 3,5,6,7,8,3′,4′-heptamethphoxyflavone, were identified from the segments of Newhall navel oranges. The accumulation of these identified PMFs was altered in response to LED and UV light irradiation (Figure 2). In spite of light irradiation, the contents of all PMFs identified in this study significantly decreased during the irradiation period, but red and blue light maintained higher levels of PMFs compared to other lights. After removal from light irradiation, the accumulation of PMFs continued to decrease in fruit treated with red and blue irradiation, and darkness, while their content was significantly induced by white light, UVB, and UVC irradiation, with white light irradiation exhibiting the optimal effect followed by UVB (Figure 2).

Twelve flavonoid glycosides were identified in the segments of Newhall navel oranges, specifically five flavone-*O*/*C*-glycosides (apigenin-6,8-di-*C*-glucoside, diosmetin-6,8-di-*C*-glucoside, rhoifolin-4′-*O*-glucoside, chysoeriol-6,8-di-*C*-glucoside, diosmin) and seven flavanone-*O*-glycosides (hesperidin, didymin, neoeriocitrin, narirutin, neoeriocitrin, narirutin-4′-glucoside, eriocitrin). Among these compounds, two flavone glycosides (diosmin, diosmetin 6,8-di-*C*-glucoside) and four flavanone glycosides (hesperidin, didymin, neoeriocitrin, narirutin) were increased in response to red light irradiation throughout the entire experimental period (Figure 3a). Similarly, narirutin, neoeriocitrin and didymin were also induced by UVA, UVB and UVC in spite of irradiation or removal from treatment. In the dark, the accumulation of diosmin, diosmetin 6,8-di-*C*-glucoside, hesperidin, didymin, and narirutin continued to decrease during the irradiation period, but consistently increased when the fruits were moved to natural light (Figure 3a). In contrast, the six flavonoids in Figure 3a were induced by blue light during the irradiation period, but their content rapidly decreased when removed from blue light, except for narirutin. In Figure 3b, UVA, UVB, UVC, darkness, and red light induced the accumulation of vicenin-2, rhoifolin-4′-*O*-glucoside, and narirutin-4′-glucoside throughout the entire experimental period; UVC exhibited the greatest inducing effect, and white light irradiation promoted lower levels of these compounds. However, even blue light irradiation induced the accumulation of the four flavonoids, and this effect disappeared when blue light was removed (Figure 3b). In addition, eriocitrin and neohesperidin were significantly induced by red light and UVB throughout the entire experimental period, with UVC and white light resulting in remarkable decreases in two flavonoids during the irradiation period and inducing a sharp increase when these two lights were removed. Fruit irradiated with blue light exhibited an inverse response (Figure 3c).

Three limonoids were identified, namely, limonin, epilimonin, and 7α-limonyl acetate, all of which exhibited patterns of increasing accumulation when the fruits were treated with UVA, UVB, UVC, and red and white light. In contrast, the content of limonin decreased when these forms of irradiation were removed, but the content of epilimonin in red light- and UVB-treated fruits and 7α-limonyl acetate in UVA- and UVB-treated fruits increased when irradiation was removed. Furthermore, the content of these three identified limonoids decreased during treatment with blue light and darkness but increased when irradiation was removed (Figure 4).

### 2.3. Principal Component Analysis of Flavonoid and Limonoid Responses

Principal component analysis (PCA) was performed based on the MS response value of 21 flavonoids and limonoids identified in response to different light treatments (UVC, UVB, UVC, red, blue, white light, and darkness) and storage times (0, 6, 15 days). As shown in Figure 5, the variance explained by PC1, PC2, and PC3 was 39.8, 29.1, and 9.4%, respectively. The identified compounds were clearly distinguished according to exposure to different sources of light irradiation. Polymethoxylated flavonoids (PMFs) were clustered with white light treatment sampled at storage day 15 (WL-15 day). Flavonoid glycosides were clustered with red and UVC light treatment sampled at storage day 15 (RL-15 day, UVC-15 day), and limonoids were clustered with UVC treatment at storage day 6 (UVC-6 day). Taken together, the results of the principal component analysis are in agreement with our observations of MS response.

## 3. Discussion

Light irradiation has been developed as an efficient method with no pollution to extend the shelf-life and maintain the quality of vegetables and fruits. In general, visible and ultraviolet light are both intrinsic parts of sunlight, while in the natural environment, all solar UVC (<280 nm) and most UVB (280–315 nm) rays are absorbed by the stratospheric ozone layer before reaching the Earth’s surface [37]. Artificial light, especially LED light that can be set to any wavelength, and is widely used in research on supplementary irradiation of various plant species. In recent years, the majority of studies on fruits treated with light irradiation have reported increases in total flavonoid content, but few studies have focused on the effect of light on individual flavonoids. For most studies, only one monochromatic light was used, while almost no studies have reported the effect of multiple light irradiation sources alone on flavonoids. In addition, all studies only investigated the effect of irradiation period. Few available studies concerned the delay effects when irradiation was removed. For citrus fruits, a relevant study reported that blue light (450 nm, 630 μmol/m^2^·s) induces some phenylpropanoids, such as scoparone [38], and, as reported previously, naringin and tangeretin are induced by UVC irradiation (254 nm, 0.1 W/m^2^) [39]. In this study, we investigated the effect of post-harvest light irradiation on the accumulation of individual flavonoids and limonoids in the segments of Newhall navel orange simultaneously, using five kinds of LEDs and ultraviolet light within the wavelength range of 100–660 nm, as well as one type of composite white light (100 μm/m^2^·s) and darkness, to identify the optimal light treatment for maintaining and improving health-promoting components in citrus fruits.

Herein, we detected 21 phytochemicals in the segments of Newhall navel oranges (*Citrus sinensis* Osbeck) and further identified six PMFs, twelve flavonoid glycosides, and three limonoids, which are almost identical to citrus characteristic compounds reported previously [40,41,42]. Previous studies demonstrated that red and blue light treatment increases flavonoid content in microgreens [43]. Blue light, rather than red light, enhances the accumulation of flavonoids in callus cultures of *Saussurea* [44]. In the present study, we found that even red and blue light could not induce the accumulation of PMFs in the segments of Newhall navel oranges. However, red and blue light maintained higher levels of PMFs compared to other light sources during the irradiation period. More importantly, post-harvest white light (100 μm/m^2^·s) irradiation specifically enhanced the accumulation of PMFs when light was removed, suggesting that each compound responds differently to a single light source. In *Arabidopsis thaliana*, key enzymes of anthocyanin biosynthesis are strongly induced by blue light, and the induction intensity decreases in the order of UVA, UVB, and red light [45]. In this study, despite the content of twelve flavonoid glycosides all obviously increasing during blue light irradiation, these contents were dramatically decreased after removal of the light. Unlike blue light, red light significantly increased the content of diosmetin-6,8-di-*C*-glucoside, diosmin, hesperidin, neoeriocitrin, narirutin, and didymin. Furthermore, UVC remarkably elevated levels of vicenin-2, rhoifolin-4′-*O*-glucoside, narirutin-4′-glucoside, and stellarin-2 in fruit segments, and the accumulation effects remained after removal of the light source until the last day of observation, which was on day 15 of storage. Therefore, red light and UVC irradiation might represent a more efficient method for improving flavonoid content after harvest, which is an essential demand of post-harvest logistic systems of fresh fruits.

Compared to flavonoids, little information is available concerning the effect of light irradiation on the accumulation of limonoids. To our knowledge, the first relevant study was reported by Jairam et al. and demonstrated that limonin and nomilin were reduced in irradiated (300 gray) freeze-dried grapefruit pulp [46]. Our results show that only UVC irradiation induced limonin, epilimonin, and 7α-limonyl acetate during six days of continuous irradiation, despite the content decreasing again after removal of the light. Similar effects were observed in response to other light treatments. Although limonoids were first discovered as bitter citrus components, their rich biological activities were later found to promote human health, which is undoubtedly more important [40]. The above results demonstrated that both LED and UV light affect the metabolism of flavonoids and limonoids in citrus fruits.

A previous report found that irradiation of pear with UVB (280–315 nm) and/or fluorescent lamps for 10 days after harvest not only results in a significant increase in the anthocyanin, flavonoids, chlorophyll, and carotenoid concentration in the peel, but also leads to a slight increase in soluble solid and organic acid concentration and a significant increase in total sugars in the flesh [15], indicating that post-harvest irradiation can permeate through the peel of the whole fruit to affect the inner pulp. Our present study provides further evidence for this. Even so, besides the light quality, the intensity of light is another important factor affecting the accumulation of metabolites in fruits, especially in the inner part of the flesh. Therefore, when a specific intensity of light was used to irradiate fruit, the intensity of light may be different for each part of the fruit. For this point, further work is required to screen for the optional intensity of light for flesh.

Light-controlled flavonoid biosynthesis in fruits is known to be regulated by *R2R3 MYB* transcription factors. Constitutive photomorphogenic 1 (COP1) acts as a central repressor in the light signaling pathway by interacting directly with photoreceptors to mediate flavonoid biosynthesis [47]. In visible light, the function of COP1 was repressed by interacting with activated photoreceptors, and COP1 is subsequently exported from the nucleus, allowing nuclear-localized transcription factors to accumulate and induce gene expression in light-regulated processes. The expression of structural flavonoid genes is directly regulated by *R2R3 MYB* transcription factors which may be regulated by basic leucine zipper (bZIP) transcription factors such as *HY5*. Ultraviolet B radiation is strongly absorbed by tryptophan (Trp) amino acid residues in the dimeric form of a UV resistance locus 8 (UVR8) photoreceptor leading to the monomerization of UVR8. Monomeric UVR8 and COP1 form a complex that accumulates in the nucleus of the cells [48,49]. The UVR8–COP1–SPA complex stabilizes the bZIP transcription factor *HY5*, promoting the activity of different R2R3 MYBs for the transcription of specific flavonoid biosynthesis genes. However, the possible mechanism of limonoids in response to light irradiation needs to be further studied in the future.

## 4. Materials and Methods

### 4.1. Chemicals

The HPLC-grade acetonitrile and methanol were purchased from Sigma–Aldrich (St. Louis, MO, USA). The HPLC-grade formic acid was purchased from Fisher Scientific (Shanghai, China). Ultrapure water was acquired using the Millipore water purification System (Bedford, MA, USA).

### 4.2. Sample Treatment

Newhall navel oranges were collected in December 2018 from the National Citrus Germplasm Repository, Citrus Research Institute of Chinese Academy of Agricultural Sciences, Chongqing, China. Ten fruit trees with the same age and similar growth condition were marked at the same orchard before sampling. Fully ripe fruits were picked randomly from four directions on these marked trees. Then, the fruits were randomly divided into 7 groups, and 180 fruits were included in each group, and these were then treated with different wavelengths of light. For each sampling, 60 fruits were taken out, 20 fruits each as biological replicates. Treatment conditions were set using the RXZ-300D intelligent growth cabinet with adjustable light irradiation parameters, purchased from Southeast Instrument Company (Ningbo, China), and all the UV TL-D36/16 lamps were purchased from Philips Company (Netherland), and the LED RDN-500B-C lamps were purchased from Ningbo Southeast Instrument Co., Ltd. (Ningbo, China). The control group was stored at 20 °C and 90–95% relative humidity (RH) in the dark. The other groups were irradiated with UVC (10–280 nm, 100 μm/m^2^·s), UVB (270–315 nm, 100 μm/m^2^·s), UVA (315–400 nm, 100 μm/m^2^·s), blue light (470 nm, 200 μm/m^2^·s), red light (660 nm, 150 μm/m^2^·s), and white light (100 μm/m^2^·s) at 20 °C and 90–95% relative humidity (RH) for 6 days. On day 6, all lights were removed, while the same temperature and relative humidity conditions were maintained for 10 days. Fruits from each group were sampled at 0, 6, and 15 days. Sample fruits were peeled to retain only the edible portion (segments) and were immediately frozen in liquid nitrogen, powdered, and stored at −80 °C until analysis.

### 4.3. Extraction of Flavonoids and Limonoids

Flavonoids and limonoids were crude extracted by methanol according to the method of Xing et al. [50] with slight modifications. Four grams of segment powder from oranges were fully mixed with 7 mL methanol. At room temperature, mixtures were placed in a 300-W ultrasonic bath for 30 min and centrifuged at 5000 rpm for 20 min. The supernatant was recovered in a 25-mL brown volumetric flask. The precipitate was again extracted twice using the same method. All 21-mL mixed supernatants were finally diluted using methanol to 25 mL in brown volumetric flasks. Methanol extracts were filtered using 13-mm syringe filters with 0.22-μm PTFE membranes before passing through the UPLC column.

### 4.4. UPLC-qTOF-MS Analysis and MS Response Quantification

Samples were analyzed using UPLC/Xevo G2-S Q-TOF MS (Waters MS Technologies, Manchester, U.K.), equipped with an ACQUITY UPLC BEH C18 column (2.1 × 100 mm, 1.7 mm, UK) coupled with the UNIFI Platform in triplicate. The analysis method was performed according to Zhao et al. [51]. Briefly, the column heater was set at 40 °C and 1 μL of each sample was injected and eluted with 0.1% formic acid in water (solvent A) and acetonitrile (solvent B) with a linear gradient (0–3 min, 5–15% B; 3–8 min, 15–25% B; 8–9 min, 25–35% B; 9–13 min, 35–45% B; 13–15 min, 45–60% B; 15–16 min, 60–90% B; 16–18 min, 90–90% B; 18–20 min, 90−5% B) at a flow rate of 0.4 mL/min. Mass spectrometry was performed in the positive ionization mode in both low- (6 V) and high-energy (20–40 V) conditions. Mass spectrum conditions were as follows: source temperature of 120 °C, desolvation temperature of 400 °C, capillary voltage of 1000 V, cone voltage of 40 V, cone gas flow of 50 L/h and desolvation gas flow rate of 800 L/h. Leucine–enkephalin (*m*/*z* 556.2766) was used as the lock-mass compound at a concentration of 200 ng/mL and flow rate of 10 μL/min. Mass spectrometry data were collected between *m*/*z* 100 and 1200. The identification and quantification of target compounds were performed according to the previous methods [50,51]. The target compounds were identified by the retention time, quasi-molecular ion, molecular formula, and dominant or diagnostic MS^2^ ion, and MS responses represent the relative content of the compounds.

### 4.5. Statistical Analysis

The peak area of the quasi-molecular ion for each identified compound was integrally computed using UNIFI software (Waters, Milford, MA, USA). Statistical differences of compound contents for each treatment point were evaluated by Fisher’s protected least squares difference (LSD) test at a 0.05 probability. Data were expressed as the mean of three biological replicates ± standard deviation (SD), using IBM SPSS Statistics software v.23 for analyses. To determine all the variables attributed to differentiate different samples, a PCA model was developed using the contents of all target compounds as the independent variables and the treatment times as the dependent variables, respectively. Three line charts and principal component analysis (PCA) were plotted by Origin Pro 2018 (Origin Lab, Northampton, MA, USA).

## 5. Conclusions

Twenty-one flavonoids were identified in the segments of Newhall navel oranges, specifically six PMFs, five flavone glycosides, seven flavanone glycosides, and three limonoids. The accumulation of these compounds was altered in response to exposure to different LED and UV light irradiation. The levels of PMFs decreased significantly during the irradiation period, and red and blue light maintained higher levels of PMFs. White light and UVB significantly induced the accumulation of PMFs after the fruits were removed from irradiation. Red light enhanced the content of diosmetin 6,8-di-*C*-glucoside, diosmin, hesperidin, neoeriocitrin, narirutin, and didymin throughout the entire experimental period with light irradiation, whereas UVC enhanced the accumulation of vicenin-2, rhoifolin-4′-*O*-glucoside, narirutin-4′-glucoside, and stellarin-2. Three limonoids, limonin, epilimonin, and 7α-limonyl acetate, were induced by UVC during the irradiation period but decreased when the irradiation was removed. These results suggest that single light sources, such as red light and UVC, have great potential for improving the nutrition and flavor of citrus fruits, which can be used for post-harvest logistic systems and lighting conditions for supermarket sales. However, the specific molecular mechanisms underlying these effects require further investigation.

## Figures and Tables

**Figure 1 molecules-24-01755-f001:**
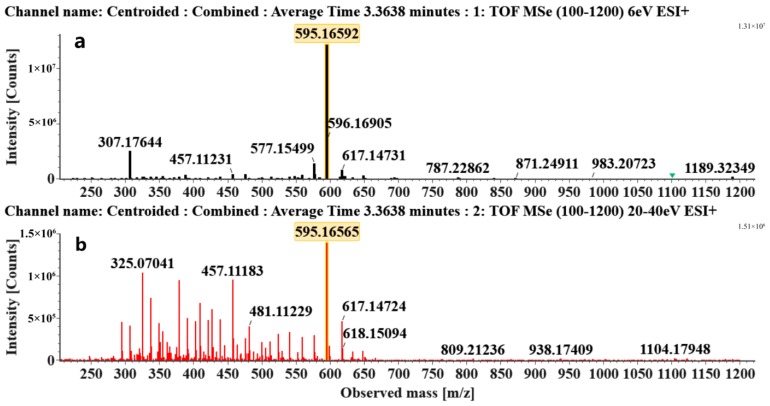
Mass chromatogram of apigenin-6,8-di-*C*-glucoside (Vicenin-2) in positive (**a**) low-energy and (**b**) high-energy modes.

**Figure 2 molecules-24-01755-f002:**
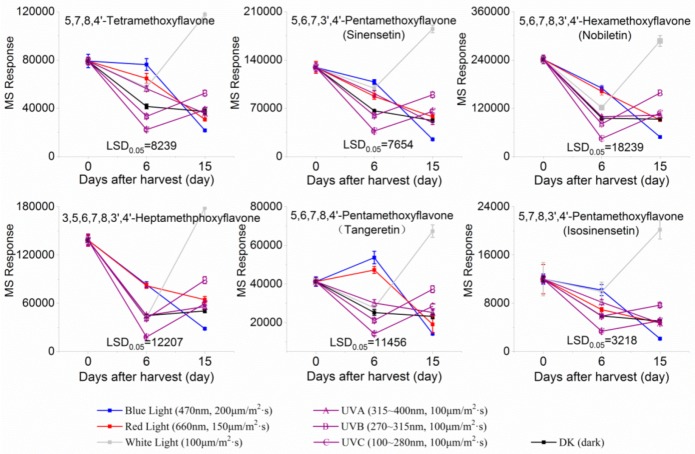
MS response of PMFs in the segments of Newhall navel orange during light irradiation and the removal period.

**Figure 3 molecules-24-01755-f003:**
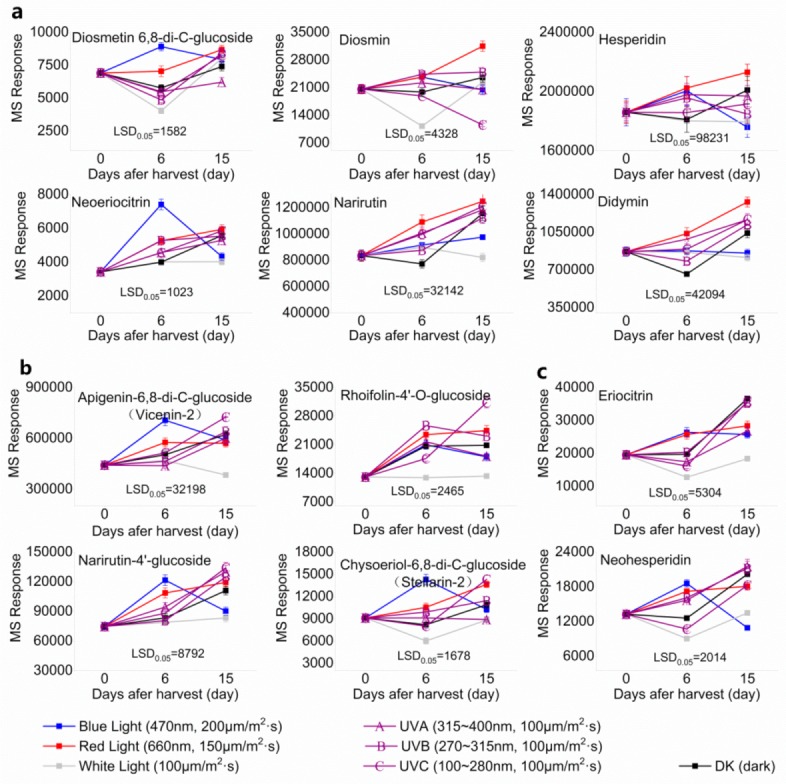
MS response of flavone and flavanone in the segments of Newhall navel orange during light irradiation and during the removal period. Flavone and flavanone (**a**) were mostly induced by red light, (**b**) UVC, and were also (**c**) induced by UV and darkness.

**Figure 4 molecules-24-01755-f004:**
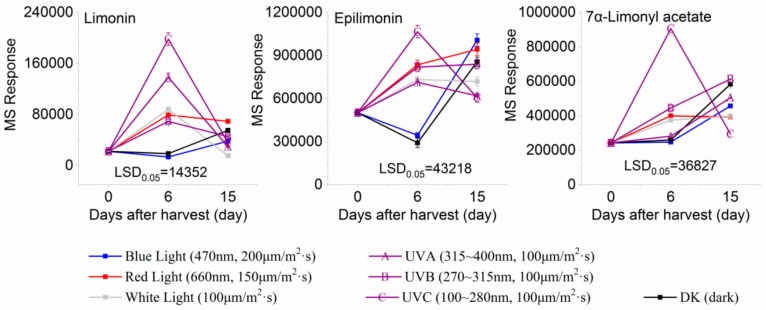
MS response of limonoids in the segments of Newhall navel orange during light irradiation and during the removal period.

**Figure 5 molecules-24-01755-f005:**
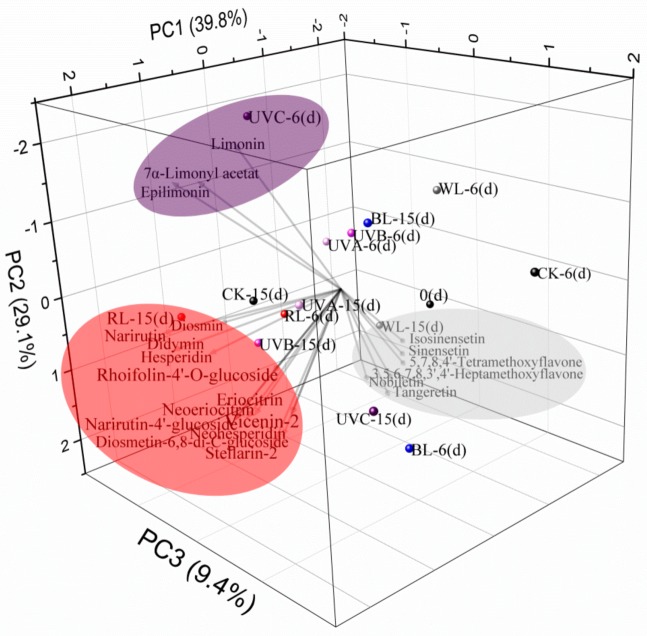
PCA model based on 21 phytochemicals in Newhall navel orange in response to different forms of light irradiation. DK: Dark; WL: White light; BL: Blue light; RL: Red light; UVA: Ultraviolet A; UVB: Ultraviolet B; UVC: Ultraviolet C; d: Days after harvest.

**Table 1 molecules-24-01755-t001:** Summary of the 21 compounds identified.

Retention Time (min)	Component Name	Formula	[M + H]^+^ (Error, ppm)	Diagnostic MS^2^ Ion (%)	Structure	CAS Registry Number	Health-Promoting Properties
3.36	Apigenin-6,8-di-*C*-glucoside (Vicenin-2) (Flavone *C*-glycoside)	C_27_H_30_O_15_	595.16592 (0.29)	595.16565 (100), 325.07041 (74.91), 457.11183 (68.54), 379.08087 (67.98), 409.09118 (48.64), 477.11946 (2.90)	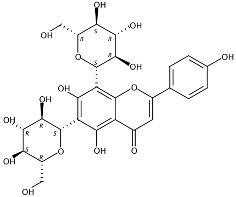	23666-13-9	Anti-prostate cancer [22]
3.59	Diosmetin 6,8-di-*C*-glucoside (Flavone *C*-glycoside)	C_28_H_32_O_16_	625.17556 (−1.20)	625.17448 (100), 355.07895 (97.31), 487.12160 (76.16), 409.09206 (75.73), 457.11214 (63.68), 367.08299 (55.83), 607.16286 (25.34)	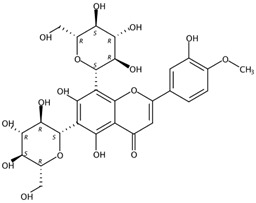	98813-28-6	Antioxidant; Antihypertensive effect [23]
3.69	Rhoifolin-4′-*O*-glucoside(Flavone *O*-glycoside)	C_33_H_40_O_19_	741.22360 (−0.07)	433.11258 (71.19), 271.05979 (68.12), 595.16594 (59.59), 153.01818 (9.23), 163.03854 (6.68)	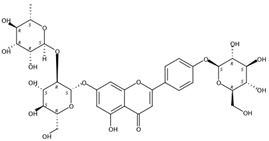	31498-83-6	Not found
3.71	Neoeriocitrin(Flavanone *O*-glycoside)	C_27_H_32_O_15_	597.17996 (−2.41)	289.07002 (35.39), 153.01818 (9.23), 435.12045 (6.36), 451.12259 (4.27)	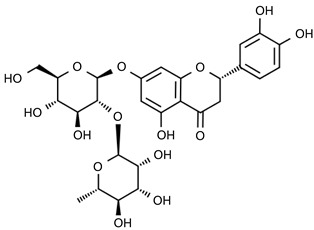	13241-32-2	Anti-osteoporosis [24]
3.76	Chysoeriol-6,8-di-*C*-glucoside(Stellarin-2) (Flavone *C*-glycoside)	C_28_H_32_O_16_	625.17571 (−0.96)	285.07510 (3.29), 457.10814 (3.07), 355.07804 (2.69), 367.07961 (1.65), 487.12086 (1.53), 607.16347 (0.48)	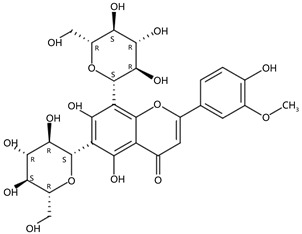	63975-58-6	DNA-binding activity [25]
3.78	Narirutin-4′-glucoside(Flavanone *O*-glycoside)	C_33_H_42_O_19_	743.23932 (−0.02)	273.07568 (100), 765.22083 (35.37), 153.01801 (18.88), 147.04375 (6.36), 435.12831 (18.31)	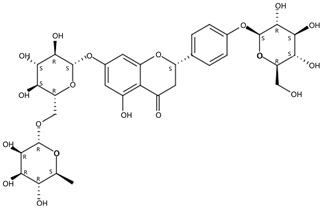	17257-22-6	Antioxidant [26]
4.34	Epilimonin(Limonoid)	C_26_H_30_O_8_	471.20201 (1.41)	471.20151(100), 425.19591 (75.41), 161.05961 (32.74), 409.20043 (12.39), 315.15814 (3.13), 273.12685 (1.79)	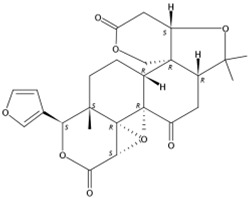	1180-71-8	Not found
4.68	Eriocitrin(Flavanone *O*-glycoside)	C_27_H_32_O_15_	597.18124 (−0.26)	289.07048 (100), 153.01781 (34.16), 163.03866 (19.72), 417.12444 (7.56), 435.12720 (7.19)	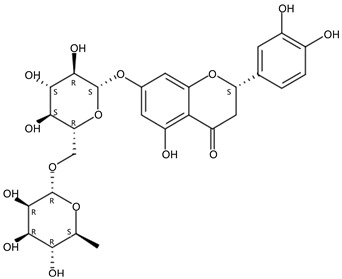	13463-28-0	Anti-obesity [27]; Reducing blood fat [28]
5.65	Narirutin(Flavanone *O*-glycoside)	C_27_H_32_O_14_	581.18661 (0.22)	273.07588 (100), 329.15998 (0.04), 153.01794 (29.66), 493.04405 (0.49), 419.13333 (18.54)	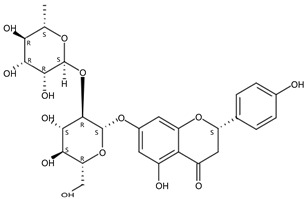	15822-82-9	Anti-inflammation [29];
6.26	Diosmin(Flavone *O*-glycoside)	C_28_H_32_O_15_	609.18126 (−0.22)	301.07027(100), 463.12343(18.13), 286.04676(11.08), 258.05273(10.79)	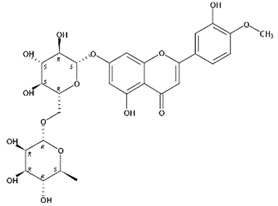	520-27-4	Antidiabetic [24]
6.41	Hesperidin(Flavanone *O*-glycoside)	C_28_H_34_O_15_	611.19777 (1.18)	303.08637 (100), 449.14389 (27.63), 153.01803 (23.49), 177.05451 (17.19), 465.13864 (13.85)	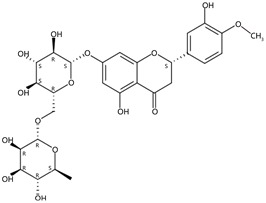	520-26-3	Antioxidant; Anti-inflammation; Anticancer; Anti-atherosclerotic effects; Vasodilatation effects [20]
6.75	Neohesperidin(Flavanone *O*-glycoside)	C_28_H_34_O_15_	611.19631 (−1.20)	303.086432 (100), 359.01192 (4.22), 153.01814 (41.50), 345.10040 (9.97)	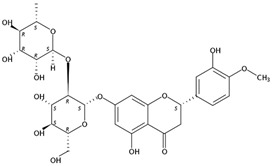	13241-33-3	Anti-gastric disease [30]
6.85	7α-Limonyl acetate(Limonoid)	C_28_H_34_O_9_	515.22775 (0.37)	515.22780 (100), 161.05954 (22.10), 303.08669 (2.38), 487.23253 (28.24), 469.22182 (17.67)	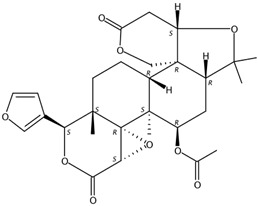	1110-03-8	Hsp90 inhibition activity [31]
8.99	Didymin(Flavanone *O*-glycoside)	C_28_H_34_O_14_	595.20210 (−0.05)	287.09139 (100), 153.01808 (29.95), 389.12180 (0.26), 161.05957 (12.80), 433.14904 (16.67)	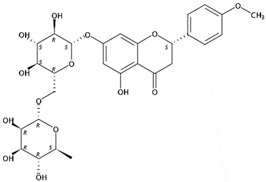	14259-47-3	Anticancer [32]; Antioxidant; Anti-inflammation; prevention of cardiovascular complications in diabetes [33]
10.41	5,7,8,3′,4′-Pentamethoxyflavone(Isosinensetin) (Polymethoxylated flavone)	C_20_H_20_O_7_	373.12780 (−1.01)	343.08079 (100), 373.12755 (41.38), 315.08636 (19.74), 357.09485 (15.26), 153.06825 (11.84), 181.08895 (8.36)	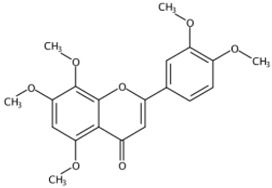	17290-70-9	Anticancer [34]
11.06	5,6,7,3′,4′-Pentamethoxyflavone(Sinensetin) (Polymethoxylated flavone)	C_20_H_20_O_7_	373.12819 (0.02)	343.08115 (100), 373.12778 (86.23), 312.09888 (58.65), 358.10324 (25.10), 153.01835 (9.06), 163.07473 (5.57)	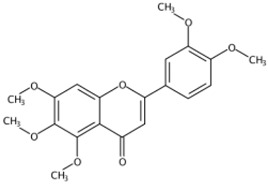	2306-27-6	Anti-inflammation; Anticancer [20]
11.18	Limonin(Limonoid)	C_26_H_30_O_8_	471.20110 (−0.51)	343.08084 (100), 328.05373 (6.03), 161.06004 (7.74), 395.10636 (8.58), 425.19653 (6.58)	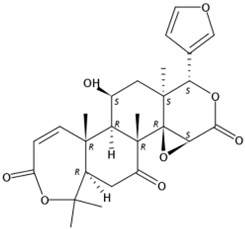	23885-43-0	Inhibit HIV-1 replication [35]; Anticancer [36]
11.95	5,6,7,8,3′,4′-Hexamethoxyflavone(Nobiletin) (Polymethoxylated flavone)	C_21_H_22_O_8_	403.13818 (−1.40)	373.09128 (100), 403.13793 (30.35), 388.11432 (12.31), 327.08555 (10.22), 211.02312 (4.26), 183.02844 (3.77)	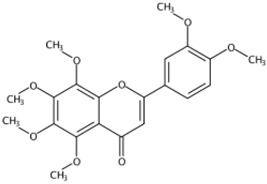	478-01-3	Anti-inflammation; Anticancer; Anti-atherosclerotic effects; Anti-diabetic effects [20]
12.03	5,7,8,4’-Tetramethoxyflavone(Polymethoxylated flavone)	C_19_H_18_O_6_	343.11751 (−0.30)	313.07010 (100), 282.08813 (68.95), 343.11731 (54.57), 153.01802 (13.31), 181.01279 (6.96), 133.06470(6.08)	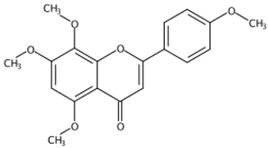	6601-66-7	Not found
12.58	3,5,6,7,8,3′,4′-Heptamethoxyflavone(Polymethoxylated flavone)	C_22_H_24_O_9_	433.14855 (−1.75)	403.10148 (100), 433.14821 (44.27), 373.05410 (5.31), 404.10479 (23.30), 418.12464 (15.15), 385.09043(10.56)	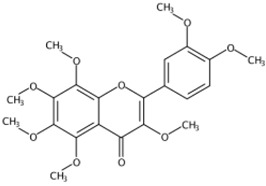	1178-24-1	Anti-inflammation; Anti-atherosclerotic effects [20]
13.02	5,6,7,8,4′-Pentamethoxyflavone(Tangeretin) (Polymethoxylated flavone)	C_20_H_20_O_7_	373.12765 (−1.42)	343.08087 (100), 395.10987 (3.85), 373.12802 (20.26), 344.08433 (20.73), 297.07575 (10.69), 211.02316 (6.07)	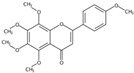	481-53-8	Antioxidant; Anti-inflammation; Anticancer; Anti-atherosclerotic effects; Anti-diabetic effects [20]; neuroprotective effects [14]

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
