# Peer review of "Effect of Post-Harvest LED and UV Light Irradiation on the Accumulation of Flavonoids and Limonoids in the Segments of Newhall Navel Oranges (Citrus sinensis Osbeck)"

_molecules, 2019, doi:10.3390/molecules24091755_

Round 1
Reviewer 1 Report
In this study, the effects of postharvest light irradiation on the accumulation of flavonoids and limonoids were investigated in the harvested Newhall navel oranges. The authors found that the contents of flavonoids and limonoids were altered after light irradiation. In fact, the regulation of flavonoid metabolism by light irradiation has already been reported in different crops. Thus, I think the research in this study is not new; the authors should point the new findings of this research, and discussed their mechanisms in the manuscript. In addition, I have several concerns that should be answered before recommending acceptance.
1. In the study, the authors compared the effects of different lights on the accumulation of flavonoids and limonoids. However, the intensities of the blue LED light and red LED light were different in this study. I think to elucidate the effects of different lights using the same intensity might be more reasonable.
2. After irradiated with different lights the fruits were peeled and only the edible parts were remained for determining the contents of flavonoids and limonoids. Why the authors did choose the edible parts as the materials in this research? Did the authors have evidences that these lights can permeate the peels of oranges?
3. I suggested that the abstract should be revised, as some sentences were difficult to understand.
Author Response
Response to Reviewer 1 Comments
Comments and Suggestions for Authors:
In this study, the effects of postharvest light irradiation on the accumulation of flavonoids and limonoids were investigated in the harvested Newhall navel oranges. The authors found that the contents of flavonoids and limonoids were altered after light irradiation. In fact, the regulation of flavonoid metabolism by light irradiation has already been reported in different crops. Thus, I think the research in this study is not new; the authors should point the new findings of this research, and discussed their mechanisms in the manuscript. In addition, I have several concerns that should be answered before recommending acceptance.
Response: Thank you for your suggestions. Thus far, LED irradiation has been extensively used to regulate plant growth and development under protection conditions, but it is also used as a postharvest measure to improve fruit quality. Yes, some studies have investigated the light irradiation on flavonoids, but these studies only focused on investigating the effect of single-color light on fruit quality and related metabolites; few publications have reported on using multiple-color light, especially both LED and UV light. Though fruits treated with light irradiation have reported increases in total flavonoid content, but few studies have focused on the effect of light on individual flavonoids. For flavonoids, only some progress has been made only in UVB treatment, but a global evaluation of other light sources regarding flavonoids and limonoids is still lacking. In addition, all studies only investigated the effect of irradiation period, few available studies concerned the delay effects when irradiation was removed. From these points, our study gave an overview of the effect of multiple light alone on main individual flavonoids and limonoids in citrus fruit during irradiation period and after irradiation was removed.
To highlight the novelty of our present study, related information was added in “Introduction” (lines 45-53) and “Discussion” (lines 179-184) in the revised manuscript.
Point 1: In the study, the authors compared the effects of different lights on the accumulation of flavonoids and limonoids. However, the intensities of the blue LED light and red LED light were different in this study. I think to elucidate the effects of different lights using the same intensity might be more reasonable.
Response 1: Thank you for pointing it out. Sure, the same intensities of the lights irradiation should be used in the similar study, which seems to be convenient to be compared. However, many studies have proved that only some specific intensity for one kind of light is efficient for postharvest treatment. Before designing the study, we investigated many previous study, and confirmed the intensity of each light (Ma et al. Journal of Agricultural and Food Chemistry 2012 60 (1), 197-201; Harbaum-Piayda et al. (2010). Postharvest Biology and Technology 56(3): 202-208; Ma, G., et al. (2015). Postharvest Biology and Technology 99(Supplement C): 99-104; Atta, M., et al. (2013). Bioresource Technology 148: 373-378.). Based on the point, different intensities of the blue LED light and red LED light, even different intensities of other light were used in the study.
Point 2: After irradiated with different lights the fruits were peeled and only the edible parts were remained for determining the contents of flavonoids and limonoids. Why the authors did choose the edible parts as the materials in this research? Did the authors have evidences that these lights can permeate the peels of oranges?
Response 2: Yes, In the study, we focus on the effect of light irradiations on flavonoids and limonoids in edible parts. Only this section is important for consume for fresh fruit. Newhall navel orange (Citrus sinensis Osbeck) is one of the most important orange for fresh citrus. Therefore, many similar studies also only choose the edible parts as the materials. In addition, it is a non-destructive detection method to be used in screening fruit with high quality in the future. Though we have not the direct evidences that these lights can permeate the peels of oranges, but we believe that the irradiation changed remarkably the light condition of edible parts. The characteristic of light permeating in different section of fruit may be investigated in the future.
Point 3: I suggested that the abstract should be revised, as some sentences were difficult to understand.
Response 3: Thank you very much. The abstract was revised, we made our effort to improve some sentences that were difficult to understand. See the new abstract of the revised manuscript, lines 19-22.
Reviewer 2 Report
The aim of the ms is very interesting regarding the use of light (UV-A/B/C and LED) to increase the bioactive compounds of oranges. The Introduction is concise and complete. Material and methods section is appropriate. The results and discussion are well presented. Conclusions are clear and summarize the main findings.
Minor comments:
· The model and brand of UV and LED lights must be included in the M&M section.
· The size of molecule structures from Table 1 must be increased.
· The statistical description (PCI, replicates, etc.) is missing in the M&M section.
Author Response
Response to Reviewer 2 Comments
Comments and Suggestions for Authors:
The aim of the ms is very interesting regarding the use of light (UV-A/B/C and LED) to increase the bioactive compounds of oranges. The Introduction is concise and complete. Material and methods section is appropriate. The results and discussion are well presented. Conclusions are clear and summarize the main findings.
Point 1: The model and brand of UV and LED lights must be included in the M&M section.
Response 1: Thanks for your suggestions. The model and brand of UV and LED lights were added to the M&M section, see lines 251-253.
Point 2: The size of molecule structures from Table 1 must be increased.
Response 2: We have increased the size of molecule structures in Table 1. See Table 1 in pages 4-10.
Point 3: The statistical description (PCI, replicates, etc.) is missing in the M&M section.
Response 3: Thanks for your pointing it out. Statistical analysis was added in the revised manuscript. In the study, three biological replicates were used for each determination. Statistical differences of compound contents for each treatment point were evaluated by Fisher's protected least square difference (LSD) test at a 0.05 probability. See M&M section, lines 288-297.
Reviewer 3 Report
Light irradiation is a green method to improve the shelf life and bioactive qualities of fruits and vegetables. The present work is very interesting and could be an important support to both food research and industries. Below some details that the authors should to consider in order to improve the manuscript quality:
Introduction
- Lines 52-61: Currently the high performance of UPLC-MS to identify and quantify bioactive substances is well known, many scientific reports show this. Then this information is not relevant for the article, I suggest remove lines 52-61
- Lines 47-51: Previous reports on light irradiation of N-n orange or other Citrus has been reported?
- Lines 62-64: Additional information about N-n orange should be presented in the introduction section:
1. Worldwide production of N-n orange
2. Who are the main producers of N-n orange?
3. What is the global market of N-n orange?
4. What bioactive properties are related with N-n orange and/or its bioactive compounds (e.g. flavonoids and limonoids)
- Light radiation (e.g. UV) has negative effects on fruits or foods exposed?
Materials and Methods
- Line 224: How much fruit was collected? Three trees were a suitable sample? according to some sampling plan? It was a representative sample?
- Line 224: How many oranges for each group?
- Line 229: Why 6 days? Why not lower or upper to 6 days? Please, explain this in the article and to present the suitable references
- Line 241: Flavonoids and limonoids were fractionated from crude extracts by mean the adsorption column? It is not clear, please explain in the manuscript
- Section 4.4: How the target compounds were identified? How the target compounds were quantified? Why the MS response is proportional to flavonoids or limonoids concentration? What ions was selected to monitor the compounds? These topics should be explained in detail in the manuscript
- Statistical analysis section?
1. How PCA was performed?
2. What were the independent and dependent variables?
3. Replicates were performed?
4. Statistical differences were established?
Results and discussion
- Line 70: The signal at 2.63 was analyzed?
- Lines 77-79: Some of these ions can be considered as diagnostic ion for vicenin-2? For each identified compound, do you have diagnostic ion reported? If this information is available, it should be used in the manuscript
- Lines 81-83: Additional information about the health-promoting properties of phytochemicals should be included in this section. I suggest to describe for each compound (or compounds group) its biological properties using suitable references
- Lines 84-86: What ion was used, the [M+H]+ ion, some daughter ion, all ions? This information is not clear in the material and methods section
- Line 100: maintained "the" higher levels
- Figure 2: For the treatment in darkness, day 6 compared to day 15 showed statistical differences? It is not clear in the article
- Figure 3: Is necessary to analyze the statistical differences and included in the results
- Lines 123-125: Figure 3b shows that for stellarin-2 the UVA, UVC and darkness treatments did not increase the MS response between 0 to 6 day
- Line 141: It is not totally true; epilimonin (red light and UVB) and 7α-limonyl acetate (UVA and UVB) increased when these forms of irradiation were removed
- Line 158: I suggest "principal component analysis are in agreement with our....", without the word "completely"
- Discussion: what is the possible molecular action mechanism related (or proposed) with the light radiation effect observed? These are very important to understand the results. Add this information in the discussion section
- Line 167: "as efficient and ???? method"
- line 182: Optimal conditions? An optimization process was not development
- line 185: Citrus sinensis Osbeck
Author Response
Response to Reviewer 3 Comments
Comments and Suggestions for Authors:
Light irradiation is a green method to improve the shelf life and bioactive qualities of fruits and vegetables. The present work is very interesting and could be an important support to both food research and industries. Below some details that the authors should to consider in order to improve the manuscript quality:
Introduction
Point 1: Lines 52-61:Currently the high performance of UPLC-MS to identify and quantify bioactive substances is well known, many scientific reports show this. Then this information is not relevant for the article, I suggest remove lines 52-61.
Response 1: Thanks for the comments. The description of high performance of UPLC-MS was removed from the manuscript, see “Introduction” section of the revised manuscript.
Point 2: Lines 47-51: Previous reports on light irradiation of N-n orange or other Citrus has been reported?
Response 2: Yes, no report on light irradiation on flavonoid accumulation in N-n orange or other Citrus has been reported.
Currently, previous studies have investigated the effect of red and blue LED light irradiation on β-Cryptoxanthin accumulation in the flavedo of Satsuma mandarin (Citrus unshiu Marc.) (Ma et al. (2012). Journal of Agricultural and Food Chemistry 60 (1):197-201.), the combination of ethylene and red LED light irradiation on carotenoid metabolism (Ma et al. (2015) Postharvest Biology and Technology 99:99-104.), and regulation role of red and blue LED light irradiation in ascorbic acid metabolism in citrus juice sacs of Satsuma mandarin (Citrus unshiu Marc.), Valencia orange (C. sinensis Osbeck), and Lisbon lemon (C. limon Burm.f.) (Zhang et al. (2015) Plant Science 233:134-142.).
In addition, though similar studies were conducted in pear, apple and mango, these studies focused on the effect of light on total flavonoids not individual flavonoids. For these studies, only one monochromatic light was used, while no studies have reported the effect of multiple light irradiation sources alone on flavonoids. In addition, all studies only investigated the effect of irradiation period, few available studies concerned the delay effects when irradiation was removed.
To present the new point of our manuscript, the related reviews were added in the Introduction (lines 45-53) and “Discussion” (lines 179-184) section.
Point 3: Lines 62-64: Additional information about N-n orange should be presented in the introduction section:
1. Worldwide production of N-n orange.
2. Who are the main producers of N-n orange?
3. What is the global market of N-n orange?
4. What bioactive properties are related with N-n orange and/or its bioactive compounds (e.g. flavonoids and limonoids).
Response 3: Thanks for the helpful suggestions. Yes, the material of importance should be introduced in the manuscript. The four points information were added in the introduction section, See lines 61-69.
Point 4: Light radiation (e.g. UV) has negative effects on fruits or foods exposed?
Response 4: UV light are widely used as a germicidal medium to reduce the food-borne microbial load in fresh food products. It has been also developed as an efficient method with no pollution to extend the shelf life and maintain the quality of vegetables and fruits. It is reported that the method can change the taste and flavour of some products. Even so, UV technology could still be a good alternative technology, instead of thermal treatment or application of antimicrobial compounds (Guerrero-Beltr·et al. (2004) Advantages and Limitations on Processing Foods by UV Light. Food Science and Technology International 10 (3):137-147.).
Materials and Methods
Point 5: Line 224: How much fruit was collected? Three trees were a suitable sample? according to some sampling plan? It was a representative sample?
Response 5: We are sorry for our wrong description. In fact, ten fruit trees with the same age and similar growth condition were marked at the same orchard before sampling. Full ripe fruits were picked randomly from four directions on these marked trees. Then, fruits were randomly divided into 7 groups, and 180 fruits were included in each group. For each sampling, 60 fruits were taken out, each 20 fruits as a replicate, and three biological replicates were used. See lines 245-249.
Point 6: Line 224: How many oranges for each group?
Response 6: 180 fruits were included in each group. lines 247.
Point 7: Line 229: Why 6 days? Why not lower or upper to 6 days? Please, explain this in the article and to present the suitable references.
Response 7: In fact, when designing the study, we referred the previous works in citrus (Ma et al. (2012). Journal of Agricultural and Food Chemistry 60 (1):197-201.). In similar studies, the treatment time is different. For example, exposing to UV-A, UV-B and visible light for 9 days in pears (Zhu et al. Scientia Horticulturae 2018, 229, 240-251.), UV-B irradiation for 10 or 60min for peach (Santin et al. Postharvest Biology and Technology 2018, 139, 127-134.).
Point 8: Line 241: Flavonoids and limonoids were fractionated from crude extracts by mean the adsorption column? It is not clear, please explain in the manuscript.
Response 8: We are sorry for our unclear description. Indeed, flavonoids and limonoids were not further fractionated from the methanol extracts. The methanol extract was directly analysed by UPLC-qTOF-MS. The description was modified, see lines 263-270.
Point 9: Section 4.4: How the target compounds were identified? How the target compounds were quantified? Why the MS response is proportional to flavonoids or limonoids concentration? What ions was selected to monitor the compounds? These topics should be explained in detail in the manuscript.
Response 9: Thanks for the comments.
In our lab, an efficient method was developed recently based on the linear relationship between content and MS Response value to analyze the PMFs in citrus (Zhao et al. Efficient analysis of phytochemical constituents in the peel of Chinese wild citrus Mangshanju (Citrus reticulata Blanco) by ultra high performance liquid chromatography–quadrupole time‐of‐flight‐mass spectrometry. J. Sep. Sci. 2018.), and 77 flavonoids and limoniods were identified from dried citrus peel using UPLC-Qtop-MS.
By using the method, we identified 21 flavonoids and limonoids in the fresh citrus pulp. Here, we identified the target compounds by the retention time, quasi-molecular ion, molecular formula, and dominant or diagnostic MS2 ion according to the previous study. And some important information was added in Table 1 and “Supplementary Materials”.
Considering the main purpose of this study is not to establish an identification method, and the detailed identification process of each compound is well described in the reference literature, we have only retold the identification process of the first compound (RT=3.36) in our manuscript (lines 78-86), and the necessary characteristic ion information of the other compounds were listed in Table 1 and also “Supplementary Materials”.
This study did not make the standard curve to calculate the absolute content of the target compounds. On one hand, in the study, we only aimed to compare the relative change level of the target compound in the samples with different treatments, so as to screen the optimal light treatment. On the other hand, in fact, we have tried to obtain all standard of target compounds, but only 11 target compounds could be obtained from the commercial companies, so it is currently difficult to obtain all compounds. Considering that half of the target compounds still could not be quantified, we finally decided to use MS response to directly represent the content of the compounds.
In our lab, Xing et al. have previously reported the successful application of MS response to identify and quantify 13 PMFs in the dried peels of citrus (Xing, et al. Fast separation and sensitive quantitation of polymethoxylated flavonoids in the peels of citrus using UPLC-Q-TOF-MS. J. Agric. Food Chem. 2017, 65, 2615.). All of their MS responses are positively correlated with the content within the linear range. Besides, even if beyond the linear range, the relationship between the compound content and the MS response also presented an upward logarithmic curve. Thus, for the same compound, the higher content must correspond to the higher MS response.
The [M+H]+ under the low energy mode was selected to monitor the compounds in the study. This information has been added in the revised manuscript, see line 90 and 289.
The information about identification and quantification of target compounds were added in lines 284-287.
Point 10: Statistical analysis section?
1. How PCA was performed?
2. What were the independent and dependent variables?
3. Replicates were performed?
4. Statistical differences were established?
Response 10: Thank you for pointing it out.
To determine all variables attribute to differentiate different samples, a PCA model was developed by using the contents of all target compounds as the independent variables and the treatment-time as the dependent variables, respectively. Three Line charts and principal component analysis (PCA) were plotted by Origin Pro 2018 (Origin Lab, Northampton, MA, USA). See lines 293-297.
Statistical differences of compound contents for each treatment point were evaluated by Fisher's protected least square difference (LSD) test at a 0.05 probability. Data were expressed as mean of three biological replicates ± standard deviation (SD), using IBM SPSS Statistics software v 23 for analyses. See lines 289-293.
Results and discussion
Point 11: Line 70: The signal at 2.63 was analyzed?
Response 11: Yes, the signal at 2.63 was analysed, but we can’t identify what it is. It has quasi-molecular ion [M+H]+ at 265.15504 m/z, MS2 ion at 145.02812 m/z, 177.05440 m/z and 117.03285.
Point 12: Lines 77-79: Some of these ions can be considered as diagnostic ion for vicenin-2? For each identified compound, do you have diagnostic ion reported? If this information is available, it should be used in the manuscript.
Response 12: Yes, all of these ion can be considered as diagnostic ion for vicenin-2 according to the reference literature. For example, the neutral loss of 120(0,2X) and 18(H2O) under the high energy mode is well known characteristic fragmentation of flavonoid-C-glucoside. The ion at 457.11183 m/z corresponded to [(M+H)-0,2X-H2O]+, indicating the compound is likely to be C-glucosyl flavonoid. Then, the ion at 477.11946 m/z corresponded to 1,3A+, indicating that there are two hydroxyl groups and two glucosides attached to A-ring, and only one hydroxyl group at B-ring and C-ring. Given that C-6 and C-8 in the flavonoid basic structure are the common C-glycosylation, the compound thus can be identified as vicenin-2.
The above identification process can be seen in detail in the reference literature, but we suppose that these reported processes should not be repeated in our manuscript. Therefore, we only provide the necessary information for the identification of the target compound, such as retention time, diagnostic ion and molecular formula (Table 1, Table S1).
We have added some details in the revised manuscript, see lines 81-85.
Point 13: Lines 81-83: Additional information about the health-promoting properties of phytochemicals should be included in this section. I suggest to describe for each compound (or compounds group) its biological properties using suitable references.
Response 13: Thanks for the helpful suggestions. We have added the health-promoting properties of each compound with references in Table 1. See pages 4-10.
Point 14: Lines 84-86: What ion was used, the [M+H]+ ion, some daughter ion, all ions? This information is not clear in the material and methods section.
Response 14: Sorry for missing this information. The [M+H]+ ion was used to integrated the MS response for each compound. We have added this information in the revised manuscript, see lines 90 and 289.
Point 15: Line 100: maintained "the" higher levels.
Response 15: Sorry for this obvious error. We have added "the" in the revised manuscript, see line 108 and 303.
Point 16: Figure 2: For the treatment in darkness, day 6 compared to day 15 showed statistical differences? It is not clear in the article.
Response 16: Yes. Statistical differences of compound contents for each treatment point were evaluated by Fisher's protected least square difference (LSD) test at a 0.05 probability. Data were expressed as mean of three biological replicates ± standard deviation (SD), using IBM SPSS Statistics software v 23 for analyses. See Figure 2 in page 11.
Point 17: Figure 3: Is necessary to analyze the statistical differences and included in the results?
Response 17: LSD test results were added in the Figure 3, see Figure 3 in page 12.
Point 18: Lines 123-125: Figure 3b shows that for stellarin-2 the UVA, UVC and darkness treatments did not increase the MS response between 0 to 6 day.
Response 18: Yes, it is wrong description, the stellarin-2 was deleted from the sentence, see line 131-132.
Point 19: Line 141: It is not totally true; epilimonin (red light and UVB) and 7α-limonyl acetate (UVA and UVB) increased when these forms of irradiation were removed.
Response 19: The wrong description was corrected, see lines 148-150.
Point 20: Line 158: I suggest "principal component analysis are in agreement with our....", without the word "completely".
Response 20: Thanks for this suggestion. We have removed the word "completely" from the sentence, see line 166.
Point 21: Discussion: what is the possible molecular action mechanism related (or proposed) with the light radiation effect observed? These are very important to understand the results. Add this information in the discussion section.
Response 21: The possible molecular action mechanism for flavonoids was added in “Discussion” (lines 224-236).
Point 22: Line 167: "as efficient and ???? method".
Response 22: Sorry for this obvious error. The "and" was removed from the sentence, see line 174.
Point 23: line 182: Optimal conditions? An optimization process was not development.
Response 23: we are sorry for our unclear word, it should be “optimal treatment”, see line 191.
Point 24: line 185: Citrus sinensis Osbeck.
Response 24: The “Citrus sinensis Osbeck” was written in italic style, see line 194.
Round 2
Reviewer 1 Report
The authors revised the abstract according to my comments, and now it is easy to understand. However, I think the responses to the first two questions were not so satisfied.
1 Why did the authors choose different intensities for each light in this study?
The authors answered:Before designing the study, we investigated many previous study, and confirmed the intensity of each light (Ma et al. Journal of Agricultural and Food Chemistry 2012 60 (1), 197-201; Harbaum-Piayda et al. (2010). Postharvest Biology and Technology 56(3): 202-208; Ma, G., et al. (2015). Postharvest Biology and Technology 99(Supplement C): 99-104; Atta, M., et al. (2013). Bioresource Technology 148: 373-378.). Based on the point, different intensities of the blue LED light and red LED light, even different intensities of other light were used in the study.
However, in the studies cited by the authors, I did not find the information on the suitable intensity of each light in citrus fruits.
2 The authors did not have direct evidence that the light can permeate through the peels. How do these lights affect the accumulation of flavonoids in the edible parts? The authors should discuss it in the discussion.
Author Response
Comments and Suggestions for Authors:
The authors revised the abstract according to my comments, and now it is easy to understand. However, I think the responses to the first two questions were not so satisfied.
Point 1: Why did the authors choose different intensities for each light in this study?
The authors answered:Before designing the study, we investigated many previous study, and confirmed the intensity of each light (Ma et al. Journal of Agricultural and Food Chemistry 2012 60 (1), 197-201; Harbaum-Piayda et al. (2010). Postharvest Biology and Technology 56(3): 202-208; Ma, G., et al. (2015). Postharvest Biology and Technology 99(Supplement C): 99-104; Atta, M., et al. (2013). Bioresource Technology 148: 373-378.). Based on the point, different intensities of the blue LED light and red LED light, even different intensities of other light were used in the study.
However, in the studies cited by the authors, I did not find the information on the suitable intensity of each light in citrus fruits.
Response 1: Thanks you for your pointing it out. We would like to further explain how light intensity was designed in this study. For the red light, Ma et al. first reported in 2012 that irradiation with red light at intensity of 50 μm/m2·s for six days was effective in enhancing carotenoid contents, especially the content of β-cryptoxanthin, while blue LED light had no significant effect on the carotenoid content in the flavedo of Satsuma mandarin, which can be seen in Figure 1 of the reference (Ma et al. (2012). Journal of Agricultural and Food Chemistry, 60 (1), 197-201.). Then, in 2015, Ma et al. increased the intensity of the red LED light to 150 μm/m2·s, and the results showed that the contents of β-cryptoxanthin, all-trans-violaxanthin, 9-cis-violaxanthin and lutein were simultaneously increased more along with the total carotenoid accumulation under red LED light, which can be seen in Figure 3 of the reference (Ma, G., et al. (2015). Postharvest Biology and Technology, 99(Supplement C): 99-104). Therefore, we chose red light of 150 μm/m2·s intensity in our present study.
For the blue light, based on Ma et al. first report in 2012, we inferred that the of intensity 50 μm/m2·s is not enough for citrus. However, we found a previous report that the growth rate of microalgae Chlorella vulgaris can be accelerated most significantly by blue light with 200 μm/m2·s compared with both 100 and 300 μm/m2·s, which can be seen in Figure 4, 6 and Table 1 of the reference (Atta, M., et al. (2013). Bioresource Technology, 148: 373-378.). Thus, we determined to set the blue light at 200 μm/m2·s.
In addition, since there are few reports on the optimal intensity of other lights, we set the intensity of all ultraviolet and white lights as 100 μm/m2·s, which is also the commonly used intensity of light irradiation treatment (Harbaum-Piayda et al. (2010). Postharvest Biology and Technology, 56(3): 202-208).
So we think your comment is a good idea for our future work, and some related discussion was also added in lines 225-234.
Point 2: The authors did not have direct evidence that the light can permeate through the peels. How do these lights affect the accumulation of flavonoids in the edible parts? The authors should discuss it in the discussion.
Response 2: Thanks for your comments to improve our manuscript. We are deeply sorry for that we are unable to provide the direct evidence that the light can permeate through the peels and affect flavonoids accumulation in the flesh, but we have found a related report that may indirectly prove this point. That is, irradiation of the entire pear fruits with UVB and/or fluorescent lamps for 10 days after harvest can not only promote the accumulation of various bioactive components in the peel (reference Figure 3), but also lead to slight increase of soluble solids and organic acids concentration and significant increase of total sugars in the flesh (reference Figure 4) (Sun, Y et al. (2014). Postharvest Biology and Technology, 91, 64-71). Although the report did not investigate whether flavonoids in the pear flesh were affected, our results do show that different light treatments lead to changes in flavonoids of citrus fruit edible parts, which is hard to explain if light hasn’t affected the flesh. Therefore, we have added the discussion in the revised manuscript, see lines 225-234.
Reviewer 3 Report
Accept after a minor english revision
Author Response
Comments and Suggestions for Authors:
Accept after a minor english revision.
Response: Thank you very much for your suggestions. We have make our efforts to improve the language of the manuscript (https://www.mdpi.com/authors/english ). In addition, we also used the editing service that MDPI provided (English Editor: Safiya Bibi; English Editing ID: english-9580).